# Improved Method for Efficient Generation of Functional Neurons from Murine Neural Progenitor Cells

**DOI:** 10.3390/cells10081894

**Published:** 2021-07-26

**Authors:** Abhinav Soni, Diana Klütsch, Xin Hu, Judith Houtman, Nicole Rund, Asako McCloskey, Jerome Mertens, Simon T. Schafer, Hayder Amin, Tomohisa Toda

**Affiliations:** 1Nuclear Architecture in Neural Plasticity and Aging, German Center for Neurodegenerative Diseases, 01307 Dresden, Germany; Abhinav.Soni@dzne.de (A.S.); judith.houtman@dzne.de (J.H.); Nicole.Rund@dzne.de (N.R.); 2Biohybrid Neuroelectronics (BIONICS), German Center for Neurodegenerative Diseases, 01307 Dresden, Germany; Diana.Kluetsch@dzne.de (D.K.); xin.hu@dzne.de (X.H.); 3Molecular and Cell Biology Laboratory, The Salk Institute for Biological Studies, 10010 North Torrey Pines Road, La Jolla, CA 92037, USA; asmccloskey@salk.edu; 4Neural Aging Laboratory, Institute of Molecular Biology, CMBI, University of Innsbruck, Technikerstr. 25, 6020 Innsbruck, Tyrol, Austria; Jerome.Mertens@uibk.ac.at; 5Laboratory of Genetics, The Salk Institute for Biological Studies, 10010 North Torrey Pines Road, La Jolla, CA 92037, USA; sschafer@salk.edu

**Keywords:** induced neurons, neuronal culture, neuronal network, neural stem cells

## Abstract

Neuronal culture was used to investigate neuronal function in physiological and pathological conditions. Despite its inevitability, primary neuronal culture remained a gold standard method that requires laborious preparation, intensive training, and animal resources. To circumvent the shortfalls of primary neuronal preparations and efficiently give rise to functional neurons, we combine a neural stem cell culture method with a direct cell type-conversion approach. The lucidity of this method enables the efficient preparation of functional neurons from mouse neural progenitor cells on demand. We demonstrate that induced neurons (NPC-iNs) by this method make synaptic connections, elicit neuronal activity-dependent cellular responses, and develop functional neuronal networks. This method will provide a concise platform for functional neuronal assessments. This indeed offers a perspective for using these characterized neuronal networks for investigating plasticity mechanisms, drug screening assays, and probing the molecular and biophysical basis of neurodevelopmental and neurodegenerative diseases.

## 1. Introduction

Neural progenitor cells (NPCs) give rise to neurons and glial cells, which constitute significant parts of our brain and underlie our cognition and behaviors. Their malfunction causes neuropsychiatric and neurological diseases [1,2,3,4,5]. To study the functionality of neurons and glia, *in vitro* culture models provide substantial contributions. Isolation of specific neural cell types and culturing them in a dish allows controlling cell-intrinsic and extrinsic factors to understand the fundamental cellular, biochemical, and physiological bases [6,7]. Although, *in vivo* cellular characteristic is not able to be predicted a priori from *in vitro* studies completely, *in vitro* models provide powerful tools to examine hypotheses on fundamental cellular properties of neurons.

To prepare functional neurons in culture, numerous studies employed primary neuronal culture by dissecting embryonic/postnatal brains [7,8], dissociating them, and growing dissociated cells in a defined medium. In this way, neuron-enriched cultures are accessible for various approaches, including cell biology, pharmacology, biochemistry, and physiology. However, to prepare primary culture neurons, (i) timed-pregnant dam needs to be prepared, (ii) animals need to be killed, (iii) intensive training from dissection to cell culture is required, and (iv) require more extensive starting materials because neurons are postmitotic. Altogether, the preparation of primary neuronal culture requires substantial time, effort, and resources.

To circumvent this hurdle and efforts, several cell biological approaches were used. Classically, neuroblastoma cell lines were used to investigate neuronal functionality [9]. Neuroblastoma cells can be expanded in a defined culture condition as much as we need, and they can be differentiated into neurons upon our requirements. However, neuroblastoma cell lines, such as neuro2A or PC12 cells, lack critical neuronal components, including the expression of synaptic proteins and axonal guidance proteins [10,11,12]. Therefore, findings in these cell lines may not fully reflect the functionality of authentic, functional neurons.

The other possibility is to differentiate neural stem/progenitor cells into neurons *in vitro*. Advances in stem cell biology now allow us to isolate embryonic and adult NPCs from the brain and maintain them in adherent cultures supported by fibroblast growth factor 2 (FGF2 or bFGF) and epidermal growth factor (EGF) [13,14,15]. However, the proportion of neuronal commitment from cultured NPCs is typically around 30% or less [15,16,17,18,19,20], and this fraction declines progressively over multiple passages, although some studies showed that the proportions of neurons derived from embryonic stem cells (ESCs) or NPCs isolated from the embryonic cortex are higher than 30% [21,22]. NPCs in culture gain gliogenic characteristics over time, which is also the case when NPCs are developed from ESCs [21,23]. Therefore, it was technically challenging to stably provide neuron-enriched cell culture from murine NPCs, which provide little enthusiasm for the usage in cell biological, biochemical, and pharmacological investigations.

After all, neuroscientists often rely on primary neuronal culture to address the functionality of developing neurons at cellular levels. To overcome this technical limitation and develop an efficient and convenient method to generate functional neurons on demand, we applied a direct conversion approach onto mouse NPCs [24,25,26,27]. Conventional methods for the isolation of NPCs from the brain and their maintenance in monolayer cultures were well-established and have strong advantages because these cultures represent very homogenous cell populations. In addition, the introduction of exogenous pro-neuronal pioneer transcription factors such as *Ascl1* and *Ngn2* by viral vectors were used for direct cellular reprograming [24,25]. However, both techniques were used independently. Here, we combined these methodologies to efficiently produce functional neurons from mouse NPCs. In a combination of stem cell biology techniques, we expanded inducible NPCs (inNPCs) and forced them to differentiate into functional neurons with high efficiency. We showed that inNPC-derived neurons (NPC-iNs) become functionally active neurons as early as 7 days after the neuronal induction of differentiation, develop synapses, and generate matured functional neuronal networks. This approach will provide a novel, accessible, and prompt platform for the neuroscience community.

## 2. Materials and Methods

### 2.1. Cell Culture and the Development of inNPCs

The mouse NPC line from the E15.5 embryonic cortex (C57/BL6) was isolated and cultured as described previously with minimal modifications in accordance with the guidance of the Government of Saxony [16]. NPCs were dissected, dissociated by a papain treatment without a percoll-based cell separation, and cultured in DMEM/F-12 supplemented with N2 and B27 (Invitrogen) in the presence of FGF2 (10 ng/mL), EGF (10 ng/mL), and heparin (5 µg/mL). pLVX-UbC-rtTA-Ngn2:2A:EGFP was a gift from Dr. Fred Gage (Addgene plasmid # 127288). Lentiviral particles harboring pLVX-UbC-rtTA-Ngn2:2A:EGFP (Titers ranged 1 × 10^6^ to 1 × 10^7^ colony forming unit, 10 µL) were prepared, as described previously [27], and applied to NPCs. Twenty-four hours after the application of lenti-viral particles, NPCs were treated with puromycin twice (0.5–1 µg/mL, Gibco), and selected NPCs were stocked as inNPCs. inNPCs were passaged every 2–3 days in the presence of FGF2, EGF, and heparin on laminin (2.5 µg/mL) and poly-D-lysine (PDL, 1 µg/mL)-coated 6 well plastic plates.

### 2.2. Induction of Neuronal Differentiation and Quantification of Cell Survival

inNPCs were detached from the plate using accutase (Stem Cell Tech, Vancouver, BC, Canada), and cell density was counted. 200,000 cells per well were plated on either laminin/poly-D-lysine or poly-DL-ornithine (PDLO, Sigma, St. Louis, MO, USA) coated 24 well plates. Upon plating, cells were resuspended with DMEM/F-12 supplemented with N2 and B27 in the presence of DOX (0.5–2 µg/mL), Rock inhibitor Y-27632 (10 nM, Stem Cell Tech), Forskolin (50 nM, AdoQ), and CHIR- 99021 (3 µM, Hoelzel Diagnostika, Cologne, Germany) (Induction media). 2–48 h after plating, culture media was exchanged into differentiation media, which is composed of 50% DMEM/F-12: 50% Neurobasal medium supplemented with N2 and B27 with vitamin A in the presence of Forskolin, TrkB receptor agonist LM22A4 (2 nM, AdoQ), and CHIR- 99021 (3 µM). Three days after induction (DAI), 0.1% FBS was supplemented in maturation media (BrainPhys Neuronal medium with 1% N2 and 1% SM1 supplement with TrkB receptor agonist LM22A4) and FBS was excluded from 5 DAI. In the case of recording for neuronal activity (MEA recording), BrainPhys medium (Stem Cell Tech) supplemented with N2, B27, and LM22A4 was used. To induce IEGs, NPC-iNs were treated with bicuculine (30 µM, Tocris) and 4-aminopyridine (4-AP, 100 µM, Abcam, Cambridge, UK).

To quantify cell survival, brightfield images and EGFP fluorescent images were captured from the center of each well on 1, 2, 5 and 7 DAI of DOX treatment. We first counted the number of cells with EGFP signals on 1 or 2 DAI, which was used as the number of induced cells. Subsequently, living neurons were counted based on morphology on 5 and 7 DAI, as described previously [28]. We occasionally observed higher numbers of EGFP^+^ cells on 5 DAI compared to 2 DAI in the condition of 0.5 µg/mL DOX treatment for 48 h, presumably due to the higher expression of EGFP at later time points. Cell survival was calculated in reference to the total number of EGFP^+^ cells induced on 1 or 2 DAI. Cells with prominent vacuoles or cell aggregates were not considered as living cells.

### 2.3. Immunocytochemistry

NPC-iNs attached on glass coverslips were fixed with 4% Paraformaldehyde (PFA) for 5 or 15 min depending on antibodies. To remove residual PFA and reduce background, NPC-iNs were washed once with phosphate-buffered saline (PBS) solution containing 0.1M glycine and twice with normal PBS for 5 min each. Following this, NPC-iNs were incubated for 60 min in a blocking solution containing 3% horse serum in PBS with 0.1% triton (PBS-T). Following blocking, NPC-iNs were incubated with primary antibodies for 60 min at room temperature and washed three times with PBS-T for 5 min each. NPC-iNs were further incubated in secondary antibodies for 60 min at room temperature, washed 3 times with PBS-T for 5 min each, mounted with Prolong Antifade mounting media. Imaging was performed using Zeiss 980 confocal microscopy. To quantify coexpression of NeuN, MAP2, GFAP, VGLUT1, and GAD67 with EGFP positive NPC-iNs, images from three independent experiments were processed using an identical protocol. EGFP expressing NPC-iNs were analyzed for co-expression of EGFP and protein of interest using FIJI image analysis software. In brief, the mean signals of background from three independent locations within the image were measured, and the mean signal plus 2 × standard deviation of the background signals were subtracted from each image. Following this step, double positive cells with two different markers were counted using the cell counter plugin.

List of antibodies used: rabbit polyclonal anti-NeuN (Millipore, Burlington, MA, USA, ABN78, 1:500), chicken polyclonal anti-MAP2 (Abcam, Ab92434, 1:2000), mouse monoclonal MAP2 (Sigma, M1406, 1:500), rabbit polyclonal anti-VGLUT1(Synaptic Systems, Goettingen, Germany, 135303, 1:500), rabbit anti-VGLUT1 (Thermo Fisher, Waltham, MA, USA, 48-2400, 1:500), mouse anti-GAD67 (Millipore, MAB5406, 1:500), rabbit polyclonal anti-GFAP(DAKO, Z0334,1:500), rabbit anti-active caspase 3 (R&D systems, Minneapolis, MN, USA, AF835, 1:1000), rat monoclonal anti-Sox2 (eBiosciences, San Diego, CA, USA, 14-9811-82, 1:1000), mouse monoclonal anti-PSD95 (Thermo Scientific, MA1-046,1:500), rabbit polyclonal Synaptophysin (Abcam, ab32594, 1:500), rabbit polyclonal anti-cFos (Synaptic systems, 226 003, 1:500), rabbit polyclonal anti-Arc (Novus Biologicals, Centennial, CO, USA, R-173-500, 1:500), and chicken polyclonal anti-GFP(Aves, Davis, CA, USA, GFP-1020, 1:2000). Secondary antibodies were obtained from Jackson ImmunoResearch.

### 2.4. RNA Isolation and Quantitative Reverse-Transcribed PCR

NPC-iNs were exposed to Tetrodotoxin (TTX) (1 uM) and incubated for 24 h. Next, they were treated with a combination of bicuculine (BIC, 30 uM) and 4-Aminopyridine (4-AP, 100 uM) for one hour. RNA from TTX treated and BIC/4-AP treated NPC-iNs were extracted using Trizol according to the manufacturer’s instruction (Invitrogen, Waltham, MA, USA, Cat#15596018). cDNA preparation was performed using iScript gDNA clear cDNA Synthesis kit (BioRad, Hercules, CA, USA) according to the instruction of the manufacturer. Quantitative PCR was performed using BioRad CFX Connect. Following primers for *cFos*, *Egr1,* and *GAPDH;*
*GAPDH-F* 5′-AGGTCGGTGTGAACGGATTTG, *GAPDH-R* 5′-TGTAGACCATGTAGTTGAGGTCA, c*Fos-F* 5′-ATCCTTGGAGCCAGTCAAGA, c*Fos-R* 5′-ATGATGCCGGAAACAAGAAG, *Erg1*-*F* 5′- AACACTTTGTGGCCTGAACC, *Egr1-R* 5′-AGGCAGAGGAAGACGATGAA. ΔΔC_t_ method was used to quantify the expression of *Fos* and *Egr1* obtained via qPCR. Expression of *GAPDH* was used as an endogenous control. The normality of the dataset was assessed via the Shapiro–Wilk test. F-test was performed to compare variance in the dataset. Statistical significance of expression levels (qPCR data) obtained for activity-dependent genes was assessed using a two-tailed One-sample *t*-test.

### 2.5. MEA Preparation, Recording, and Analysis

We sterilized MEA plates under the laminar flow UV-light for 1 h. The MEA plates were immediately coated with 100 ug/mL poly-DL-ornithine (PDLO, Sigma) and incubated overnight at 37 °C and 5% CO_2_ and 95% humidity. The next day, MEA plates were rinsed 4 times with sterile double-distilled water (DDW) and left to dry under the laminar hood before cell seeding. Then, we seeded cells in 100 µL of induction media to reach a final density of 150,000 cells on each well’s active recording area of the MEA plate and incubate at 37 °C with 5% CO_2_ and 95% humidity. 2 mL of differentiation media without DOX was then gently added to each well 2 h after plating and incubated under the same conditions. Media was exchanged the next day. Three days after the cell plating, the medium was replenished with BrainPhys-based maturation media. At DAI 5, 7, 10, 13, and 16, 50% of the medium was replaced with fresh medium. All reagents were obtained, unless indicated differently, from Life Technologies.

We performed all extracellular recordings using the Axion Maestro Edge acquisition system (Axion Biosystems, Atlanta, GA, USA). We used CytoView MEA plates composed of 6 wells, where each well integrates 8 × 8 PEDOT coated electrode grid with 50 µm electrode size. We collected 10 min of spontaneous firing activity at a 12.5 kHz sampling rate. Spike detection was performed with the adaptive threshold crossing algorithm with a threshold of 6× standard deviation using the AxIS software (Axion Biosystems, Atlanta, GA, USA). All spike trains were exported from AxIS software and were analyzed using custom-written Python scripts. The first-order statistics of network-wide mean activity parameters (number of active electrodes, MFR, MBR, ISI) were computed as previously described [29]. Functional connectivity maps and graph topology metrics were computed, as previously described [30].

### 2.6. Multisite Electrical Stimulation and Analysis of Evoked Responses

The NPC-iNs networks were electrically stimulated with the on-chip stimulating electrodes controlled with the AxIS software. Low frequency (0.2 Hz) trains of biphasic current stimuli (500 µs per phase, and 160 µA peak-to-peak amplitude) were delivered for 5 min using a random sequence for the stimulation sites. Then, evoked responses and stimulation events were isolated from the recording with a custom Python script. The PSTHs were computed from these isolated and aligned responses, as previously described [30]. The PSTHs were computed for each stimulating site by considering the lower and uppermost percentiles’ average overall evoked responses for each time bin.

### 2.7. Statistical Analyses

All statistical analyses were performed with Originlab 2020, R, Graph pad Prism 8. Data are expressed as the mean ± standard deviation (s.d.) or standard error margin (s.e.m) as indicated. Differences between groups were examined for statistical significance, where appropriate, using *t*-test, one-way analysis of variance (ANOVA), or two-way ANOVA followed by Tukey posthoc test.

## 3. Results

### 3.1. Development of inNPCs and Optimization of Their Differentiation

NPCs isolated from the E15.5 embryonic mouse neocortex are maintained in the presence of FGF2 and EGF [15,16]. We applied the all-in-one lentivirus vector harboring a *Neurogenin2* (*NGN2*) and EGFP under the control of tetracycline-dependent inducible promoter with rtTA under the control of UbiquitinC promoter (pLVX-UbC-rtTA-Ngn2:2A:EGFP) [27]. With this inducible vector, we designed a protocol to generate functional neurons on-demand from mouse NPCs. NPCs transduced with this lentivirus were selected with puromycin, and then they were propagated in the presence of FGF2, EGF, and heparin, which resulted as inNPCs. To induce neuronal differentiation, doxycycline (DOX) in the final concentration of 2 µg/mL was applied for 24 h upon the passage on the Poly-DL-ornithine-coated plates because this concentration of DOX worked efficiently in our previous studies [26,27]. 24 h treatment of DOX leads to the majority of cells expressing EGFP (as illustrated in Appendix A). Five days after the induction of differentiation (5 DAI) with DOX treatment, EGFP-positive cells exhibited neuronal morphology (as illustrated in Appendix A), suggesting DOX-induced NGN2 expression directed neuronal differentiation. However, EGFP-positive cells started to die at this stage (as illustrated in Appendix A, 15 ± 5.1% survival at 5 DAI and 7 ± 1.6% survival at 7 DAI compared to 1 DAI, *p* = 0.0008 1 DAI vs. 5 DAI, *p* < 0.0001 for 1 DAI vs. 7 DAI, two way ANOVA). To avoid cell death, we first tested different durations of DOX treatment. We found that the 8 h-DOX treatment induced equivalent fractions of EGFP^+^ cells compared to the 24 h-DOX treatment (8 h; 79.3 ± 2.5%, 24 h treatment; 80.0 ± 4.6%; *p* = 0.9879 for 8 h vs. 24 h, one way ANOVA), but no significant improvement in cell survival was observed at 5 or 7 DAI (as illustrated in Appendix A, 25 ± 5.3% survival at 5 DAI and 9.5 ± 4.3% survival at 7 DAI compared to DAI 1; *p* = 0.2537 for 8 h vs. 24 h at 5 DAI; *p* = 0.9651 for 8 h vs. 24 h at 7 DAI, two way ANOVA). The 2 h-DOX treatment induced EGFP^+^ cells (2 h; 50.0 ± 7.9%) and induced neuronal morphological changes five days after the DOX treatment, suggesting the 2 h-DOX treatment is sufficient to induce NGN2 for neuronal differentiation (as illustrated in Appendix A). Moreover, the 2 h-DOX treatment significantly improved cell survival at 5 DAI and 7 DAI in comparison to the 8 h- or 24 h- DOX treatment (as illustrated in Appendix A, 51 ± 9.5% survival at 5 DAI compared to 1 DAI, *p* < 0.0001 for 2 h vs. 24 h; *p* = 0.0002 for 2 h vs. 8 h, 30.1 ± 11% survival at 7 DAI; *p* = 0.0010 for 2 h vs. 24 h; *p* = 0.0033 for 2 h vs. 8 h, two-way ANOVA). In addition to shortening the DOX treatment, we also tested reducing the DOX dosage. We induced inNPC-derived neurons (NPC-iNs) with a concentration of 0.5 µg/mL. We observed a reduced induction efficiency with a low DOX dosage of 2–24 h treatments (data not shown). To achieve higher efficiency of induction, we increased the DOX-treatment time to 48 h. We observed the 48 h-treatment with 0.5 µg/mL DOX leads to comparable induction of EGFP^+^ cells (as illustrated in Appendix A, 53.1 ± 3.7%) to the 2 h treatment with 2 µg/mL DOX (as illustrated in Appendix A, 50.0 ± 7.9%). Importantly, 0.5 µg/mL DOX with 48 h treatment also exhibited improved cell survival in comparison to that of 8 h or 24 h DOX treatment (as illustrated in Appendix A, 59.8 ± 4.8% survival at 5 DAI compared to 1 DAI; *p* < 0.0001 for 48 h-0.5 µg/mL vs. 24 h-2 µg/mL; *p* < 0.0001 for 48 h-0.5 µg/mL vs. 8 h-2 µg/mL; 43.1 ± 13.9% survival at 7 DAI; *p* < 0.0001 for 48 h-0.5 µg/mL vs. 24 h-2 µg/mL; *p* < 0.0001 for 48 h-0.5 µg/mL vs. 8 h-2 µg/mL, two-way ANOVA). Since the survival rate of NPC-iNs is slightly better in 0.5 µg/mL DOX with 48 h than the 2 h-treatment with 2 µg/mL DOX, we pursued this condition in the following experiments.

To further facilitate cellular survival, we tested two treatments. First, NPC-derived induced neurons (NPC-iNs) were treated with a low concentration of fetal bovine serum (FBS, 0.1%) on 3 DAI, which is known to facilitate neuronal survival (as illustrated in Appendix A) [31,32]. The treatment with FBS increased the survival rate of NPC-iNs at 5 DAI (as illustrated in Appendix A, 86.5 ± 19.2% survival at 5 DAI compared to 1 DAI, *p* = 0.0094 for survival, FBS vs. No FBS-CHIR at 5 DAI, two-way ANOVA). However, FBS was insufficient to rescue cell death at 7 DAI (as illustrated in Appendix A, 64.8 ± 12.4% survival at 7 DAI compared to 1 DAI, *p* = 0.1271 for survival FBS vs. No FBS-CHIR at 7 DAI, two way ANOVA). Therefore, in combination with FBS, we further tested CHIR-99021 (CHIR), an agonist of canonical Wnt signaling, which is known to facilitate neuronal differentiation and survival [33,34,35]. The treatment with CHIR significantly improved the rate of survival at 7 DAI (as illustrated in Appendix A 75.3 ± 2.6% survival at 7 DAI compared to 1 DAI, *p* = 0.0068 for survival FBS-CHIR vs. no treatment at 7 DAI, two way ANOVA). Therefore, both FBS and CHIR-99021 treatments are implemented in our NPC-iN conversion protocol (as illustrated in Figure 1B). We quantified the fraction of neurons using a neuron marker NeuN in the culture. We found that 57.5 ± 3.6% of DAPI-positive cells are NeuN-positive, suggesting that majority of the cells differentiated into neurons in this condition (as illustrated in Appendix A).

### 3.2. Characterization of NPC-iNs

We then characterized NPC-iNs using several differentiation markers using immunocytochemistry. At 7 DAI, EGFP^+^ NPC-iNs expressed the neuronal markers MAP2 and NeuN (99.6 ± 0.6% within 262 EGFP^+^ cells; and 96.1 ± 4.1% within 311 EGFP^+^ cells, respectively) (as illustrated in Figure 1C,D), and their dendritic morphology developed over time (as illustrated in Figure 1E). To investigate the proportion of GABA-ergic and glutamatergic neurons in NPC-iNs, we simultaneously immunostained with anti-GAD67, a GABA-ergic neuron marker and VGLUT1, a glutamatergic marker. Intriguingly, although we observed that some of EGFP-positive cells express GAD67, most of GAD67 positive cells were also VGLUT1 positive (as illustrated in Figure 1C,F; VGLUT1^+^ = 37 ± 7.3%, GAD67^+^ = 2.9 ± 3.5%, VGLUT1^+^GAD67^+^ = 60.1 ± 3.9%). Previous studies showed that GAD67 could be expressed in immature glutarmatergic neurons [36,37,38]. These data indicate that although NPC-iNs generated by our method become mostly glutarmatergic, NPC-iNs could be immature and express GAD67 at this stage. On the other hand, the expression of astrocytic marker glial fibrillary acidic protein GFAP (0%, within 369 EGFP^+^ cells) was negligible, suggesting that most or all EGFP^+^ cells converted into neurons, but not into astrocytes. We also characterized the fate of EGFP negative cells and found that the majority of EGFP negative cells are GFAP positive (as illustrated in Appendix A, 91.1 ± 2.6%), suggesting that most of the non-induced cells differentiate into GFAP positive astrocytes in our culture condition. In addition to neuronal marker expression, we observed spines on the dendrites of NPC-iNs as early as 7 DAI (as illustrated in Figure 2A). Also, synaptic markers on dendrites (synaptophysin, PSD-95, and VGLUT1) were enriched on their dendrites at 22 and 25 DAI (as illustrated in Figure 2B), supporting that NPC-iNs display classical *in vitro* developmental features of synaptic maturation and neuronal network formation.

To test the neuronal functionality of NPC-iNs, we subsequently examined the induction of immediate early genes (IEGs) upon neuronal stimulation. NPC-iNs were initially silenced with 1 µM tetrodotoxin (TTX) and then synaptically activated by washing out TTX and adding a BIC/4AP cocktail (30 µM bicuculine, 100 µM 4-aminopyridine) which was used to potentiate synaptic transmission and induce the expression of IEGs [39,40]. After one hour of treatment with the BIC/4AP treatment, NPC-iNs were fixed and stained against the IEGs cFos and Arc. Immunocytochemical analyses revealed that cFos and Arc levels were markedly upregulated after the treatment with BIC/4AP, suggesting that NPC-iNs were synaptically activated and were able to express IEGs (as illustrated in Figure 2C). We also collected RNA of NPC-iNs before and after the treatment of BIC/4AP and found that the mRNA levels of *Fos* and *Egr1* were significantly upregulated after the treatment of BIC/4AP (as illustrated in Figure 2D,E, *Fos* = 16.26 ± 1.65, *p* = 0.0039; *Egr1* 10.77 ± 1.42, *p* = 0.0057; one sample *t*-test). These results indicate that cellular signaling pathways for activating IEGs in response to synaptic inputs are functional in NPC-iNs, and synaptic inputs onto NPC-iNs can activate signaling pathways mediating neuronal responses [41]. Taken together, our data suggest that NPC-iNs develop cellular functionality to induce IEGs in response to external stimuli.

### 3.3. Functional Maturation of NPC-iNs Networks

Neuronal electrical activity plays a critical role in regulating the synaptic strength and shaping the maturation of the developing neuronal networks *in vivo* and *in vitro* [42]. To determine the functionality and the ability of NPC-iNs to form a self-organized complex network, obtained from the recorded extracellular firing activity between 5 and 21 DAI using a multiwell platform of microelectrode array (MEAs) (Axion Biosystems, Atlanta, GA, USA), and we analyzed their firing patterns and functional features (as illustrated in Figure 3A). We observed the first spontaneous spiking activity at 8 DAI, followed by an evident change in the firing properties and synchronous patterns during later network development stages (as illustrated in Figure 3B,C). At 14 DAI, regular network-wide activity was observed, and a higher frequency of network burst patterns was dominant at 21 DAI (as illustrated in Figure 3C). The average firing electrodes increased significantly (*p* < 0.001, one-way ANOVA) from (4.6 ± 0.6 active electrodes) at 8 DAI to full array activity (64 ± 0.6 electrodes) at 21 DAI. The mean firing rate (MFR) showed a increasing tendency before plateauing at 17 DAI. The MFR computed at 8 DAI vs. 21 DAI were 0.06 ± 0.028 Hz vs. 0.89 ± 0.19 Hz, respectively (*p* < 0.01, one-way ANOVA). Furthermore, we monitored the emergence of bursting development by characterizing the mean bursting rate (MBR) of the NPC-iNs networks, which indicated at 8 DAI no burst and only random spikes (0.17 ± 0.09 burst/min) with a significantly higher bursting rate at 21 DAI (1.13 ± 0.2 burst/min) (*p* < 0.01, one-way ANOVA) (as illustrated in Figure 3C,D). Notably, the inter-spike-intervals (ISIs) were decreased during developmental stages, indicating increased bursting rate and network behavioral patterns.

Next, we sought to evaluate the synaptic strength by assessing the evoked-network responses to electrical stimulation at a mature network stage at 21 DAI. To do so, trains of slow frequency electrical stimuli at 0.2 Hz were applied sequentially from the internal distributed stimulating electrodes (as illustrated in Figure 3F). We observed elicited network-wide spiking responses following each stimulus that can be distinguished from the rastergrams and the post-stimulus-time-histograms (PSTHs) (as illustrated in Figure 3E,G). The collective network response lasted for less than 100 ms after the stimulation, followed by a network activity that returns to the prestimulation firing state (as illustrated in Figure 3G). These results indicate that NPC-iNs networks are electrically responsive and endowed with physiological properties alike those observed in primary neuronal cultures.

The increased self-organization of spike patterns during the developmental stages of NPC-iNs networks indicates enhanced network connectivity and dynamics. To examine the functional connectivity underlying the emerging firing patterns, we quantified the pairwise firing electrode interactions during three developmental weeks using the Pearson correlation (as illustrated in Figure 4A). These functional maps demonstrated the growth-formation of functional links between integrated firing neurons following the emergence of their firing patterns. These maps were also classified and quantified with complex topological features of the hub-nodes and the rich-club nodes. We found that these metrics emerged at the early stage of development (8 DAI), and they become essential elements of self-organized networks and dynamics in later matured phases (as illustrated in Figure 4B). Further analysis of graph theory on the developmental firing activity of the NPC-iN networks was used to delineate the topological organization of the functional links based on the nodal clustering coefficient, number of edges, and the degree of links. The quantification of these metrics displayed significant values at higher matured phases of development (21 DAI vs. 8 DAI, *p* < 0.05, 0.01, 0.001, one-way ANOVA) (as illustrated in Figure 4C–E).

## 4. Discussion

Here we developed an efficient, simple, user- and resource-friendly method to convert mouse NPC into functional neurons *in vitro*. Our data indicated that NPC-iNs adopt morphological characteristics of mature neurons, including spines on their dendrites. We also showed that NPC-iNs develop functional synapses and elicit the expression of IEGs upon chemical stimulation, an essential process for neuronal plasticity. These data clearly showed that NPC-iNs differentiate into functional neurons. Furthermore, through network-wide activity recordings and spiking analyses, we demonstrated that NPC-iNs can form self-organized complex networks. In turn, this allows robust information processing already at early developmental phases as analogous to well-established murine primary neuronal cultures [43,44,45]. Altogether, our approach provides a reliable platform to generate functional neurons from murine NPCs on-demand to characterize neuronal dynamics from molecular signaling to the cellular network scale. This indeed offers a perspective for using these characterized neuronal networks for investigating plasticity mechanisms, drug screening assays, and probing the molecular and biophysical basis of neurodevelopmental and neurodegenerative diseases.

A key strength of this method is that inNPCs can be easily expanded and then converted into neurons when needed. This method enables efficient preparation of sufficient starting materials without animal breeding and sacrificing. Furthermore, this method can be applied to any NPCs isolated from animals. Thus, once NPC lines are isolated from transgenic or knockout mouse lines, one can make functional neurons *in vitro* to address their neuronal properties in cell biological, physiological, pharmacological, and biochemical assays. The methods to culture mouse NPCs are well established to maintain a homogenous population; therefore, this will save many efforts and resources, including (i) the maintenance and breeding of mouse lines, (ii) time to obtain embryos/postnatal pups, (iii) the preparation of primary culture neurons, and (iv) simultaneous genotyping.

In a previous study, retroviral vectors carrying *Nurr1/Ngn2* were used to transduce rat NPCs to generate dopaminergic neurons. The study clearly showed that an introduction of exogenous transcription factors could guide differentiation into specific neuronal subtypes from murine NPCs [46]. However, with non-inducible vector systems, one needs to prepare NPCs and viruses for each round of experimentation, which is intensive in time, costs, and workload. Ho et al. also showed a similar approach for human NPCs. They used two lentiviral vectors harboring rtTA and TetO-Ngn2-puromycin independently and induced neuronal differentiation from human NPCs into neurons. The method efficiently induced neuronal differentiation [47]. However, their method requires two lentiviral vectors and can only select cells for the transduction of one vector (TetO-Ngn2-Puromycin). Our method used the all-in-one vector including rtTA, NGN2, EGFP, and puromycin, therefore we only require to make one type of virus and can select transduced cells efficiently, expand them, and induce differentiation upon demand. Therefore, our protocol significantly reduces time, costs, and workload for scientists to efficiently generate functional neurons.

Interestingly, we found that mouse NPCs are more sensitive to DOX-induced neuronal differentiation compared to human cells. In the case of human cells, including fibroblasts and induced pluripotent cells (iPSCs), a continuous DOX treatment to induce neuronal differentiation does not cause excessive cell death [26,27]. However, in the case of mouse NPCs, a strong DOX induction resulted in cell death. We tested if DOX directly affects mNPC proliferation and cell viability by treating naïve mNPCs with different concentrations of DOX. We observed no apparent differences in mNPC proliferation (as illustrated in Appendix A). We also performed immunocytochemical staining for active-caspase 3 and found no significant differences in the fraction of active-caspase 3 positive cells with or without DOX treatment (as illustrated in Appendix A, *p* = 0.1532, One way ANOVA), suggesting that this toxicity was derived from NGN2 expression instead of direct effects of DOX on NPCs. A similar cellular vulnerability was also reported in lineage reprogramming from astrocytes [28]. In the case of reprogramming from astrocytes, cell death mainly occurs at the beginning of reprogramming [28]; thus, the timing of cell death is different from NPC-iNs. However, common mechanisms may be underlying direct reprogramming-induced cell deaths at different reprogramming/differentiation time points. In the future, it would be interesting to identify responsible mechanisms of cellular vulnerability in direct reprogramming. In addition, since rodents’ developmental processes are much faster than in humans, the strong expression of pioneer transcription factors such as NGN2 and their downstream pathways may be harmful to maturating neurons after their differentiation. To highlight this, NGN2 expression is quickly downregulated during neuronal differentiation in the physiological condition [48], and shortening DOX treatment and lowering DOX doses indeed improved the survival rate of NPC-iNs. Therefore, refining the time-course and dosages of DOX-based induction may prevent cell death in NPC-iNs. As a technical note, we noticed that a higher passage of inNPCs (over 15 passages) impaired the survival rate. Therefore, it will be essential to use younger passages of inNPC for experiments.

In this study, we provided a comprehensive characterization of the electrophysiological properties of NPC-iNs based on firing patterns recorded with MEAs and correlate with the cellular quantification obtained from PSD-95, synaptophysin, V-GLUT1, cFos, and Arc expression. Our results demonstrated (i) extracellular firing activity with almost no bursting patterns at an early stage 5 DAI observed on a small number of electrodes, became more evident at 8 DAI [43,44], and their complex dynamics gradually but significantly evolved with the network development (ii) rich repertoire of synchronous bursting pattern began at 14 DAI correlated with the increased synaptic density [49,50], and neuronal connections [51] (iii) electrically evoked network response at matured developmental phase (21 DAI) displayed a global-network modulation yielded from the functional interconnected subneuronal populations [52,53] (iv) emergence of functional connectivity of developmental spontaneous firing patterns similar with primary culture neurons [54,55]. In summary, these results demonstrated that NPC-iNs networks develop and mature cellularly and functionally as dissociated murine primary neuronal networks [43,44,56], providing in return a valuable source of generating biological materials on-demand for the neuroscience community.

In our protocol, we used NGN2 to prepare pan-neuronal culture. In the future, other transcription factors can be combined with NGN2 to generate specific subtypes of neurons [57,58,59,60,61]. Furthermore, in combination with different types of NPC-iNs with different combinations of transcription factors, we could use a bottom-up approach to recapitulate the brain’s heterogeneous nature *in vitro*.

## Figures and Tables

**Figure 1 cells-10-01894-f001:**
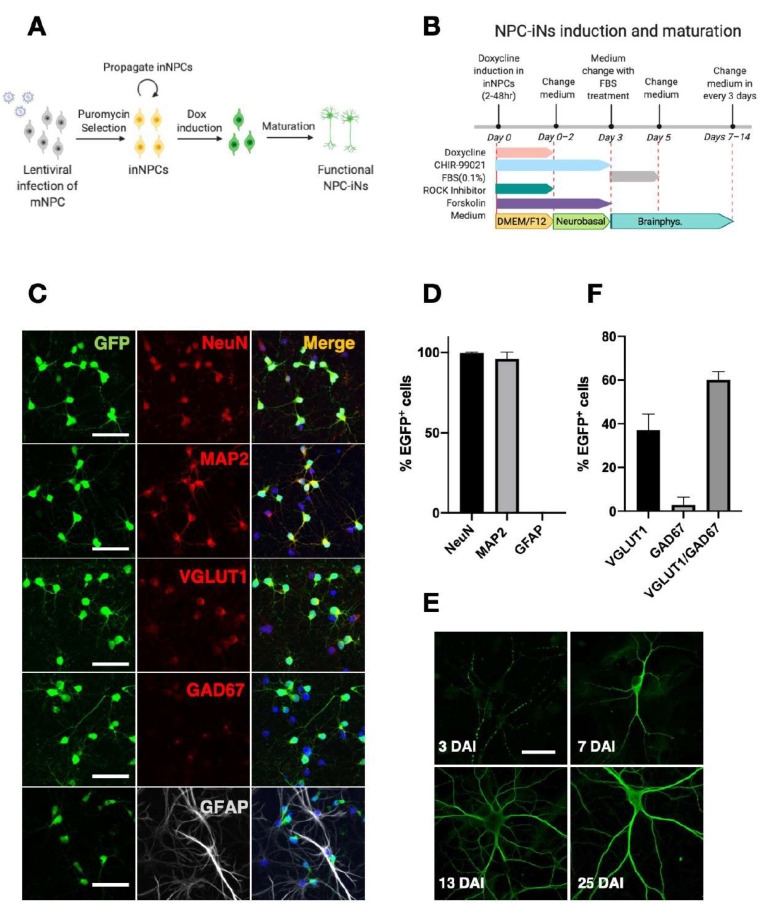
Differentiation of inNPCs into neurons. (**A**) A schema for generation of inNPCs and induction of NPC-iNs. (**B**) Experimental timeline for inducing and maturing NPC-iNs. (**C**) Characterizing differentiation of NPC-iNs at 7 DAI. EGFP^+^ NPC-iNs at 7 DAI (green) stained with neuronal markers NeuN, MAP2 (red), an excitatory neuronal marker VGLUT1 (red), a GABAergic neuronal marker GAD67 (red) and an astrocytic marker GFAP (white). Scale bars = 50 µm. (**D**) Quantification of the fraction of EGFP^+^ NPC-iNs coexpressing NeuN (N = 262 cells, 99.6 ± 0.6%), MAP2 (N = 311 cells, 96.1 ± 4.1%) or GFAP (N = 260 cells, 0 ± 0%). (**E**) Quantification of the fraction of EGFP^+^ NPC-iNs coexpressing only VGLUT1 (N = 688, 37 ± 7.3%), only GAD67 (N = 688, 2.9 ± 3.5%) and both VGLUT1 and GAD67 (N = 688, 60.1 ± 3.9%). (**F**) NPC-iNs at 3 DAI, 7 DAI, 13 DAI, and 25 DAI stained with MAP2 to show morphological development over time. Scale bar = 25 µm. All quantifications correspond to three independent experiments. Data are presented as mean ± s.d.

**Figure 2 cells-10-01894-f002:**
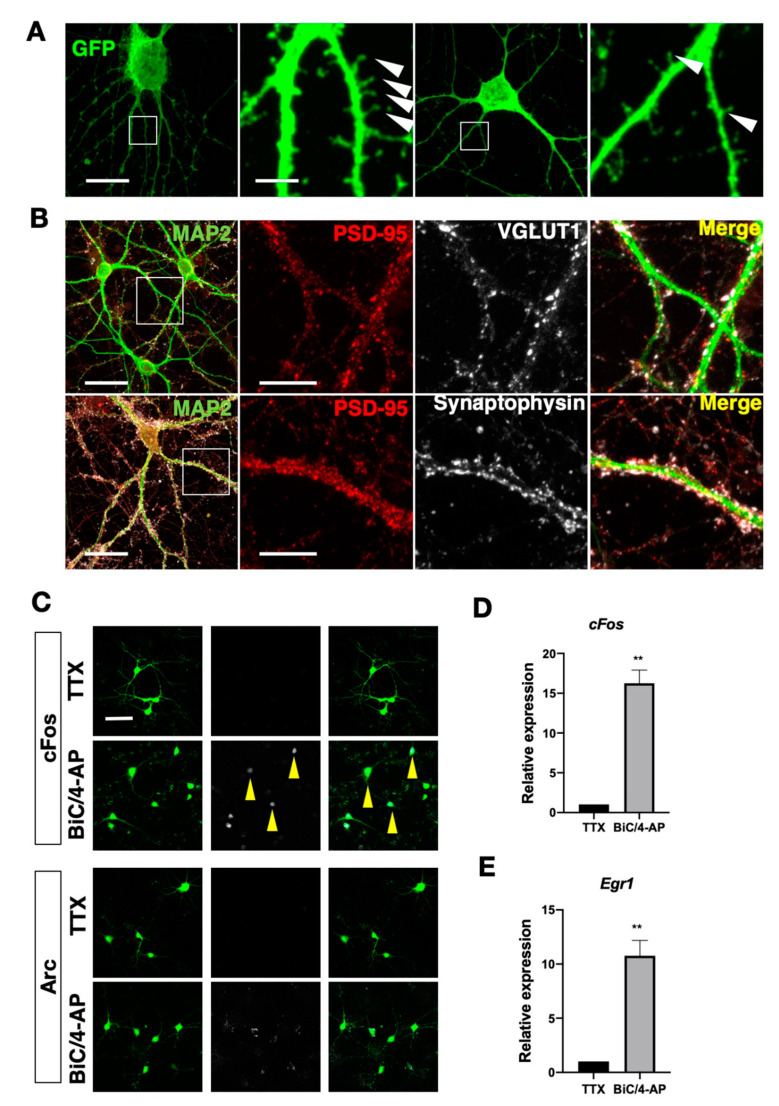
Cellular functionality and synaptic maturation of NPC-iNs. (**A**) EGFP^+^ NPC-iNs at 7 DAI (green). White square box indicates area magnified on right side, and arrowheads highlight dendritic spines. Scale bar = 25 µm (left images) and 5 µm (magnified images). (**B**) Confocal images of NPC-iNs (green) at 22 DAI (top) and 25 DAI (bottom) stained with postsynaptic marker presynaptic marker PSD95 (red), presynaptic markers synaptophysin (bottom), and VGLUT1 (top). White boxes indicate area magnified on the right side, scale bar = 28 µm (left images) and 10 µm (magnified images). (**C**) EGFP^+^ NPC-iNs at 13 DAI exposed to pharmacological agents Tetrodotoxin (TTX) or Bicuculline (BIC) and 4-Aminopyridine (4-AP), stained for immediate early genes cFos or Arc (white). Arrows highlight NPC-iNs expressing cFos or Arc. Scale bar = 100 µm. (**D**,**E**) *cFos* and *Egr1* mRNA expression levels after being exposed to TTX or BiC/4-AP. (Data for N = 3, ** *p* < 0.005, two-tailed one-sample *t*-test). Data are presented as mean ± s.d.

**Figure 3 cells-10-01894-f003:**
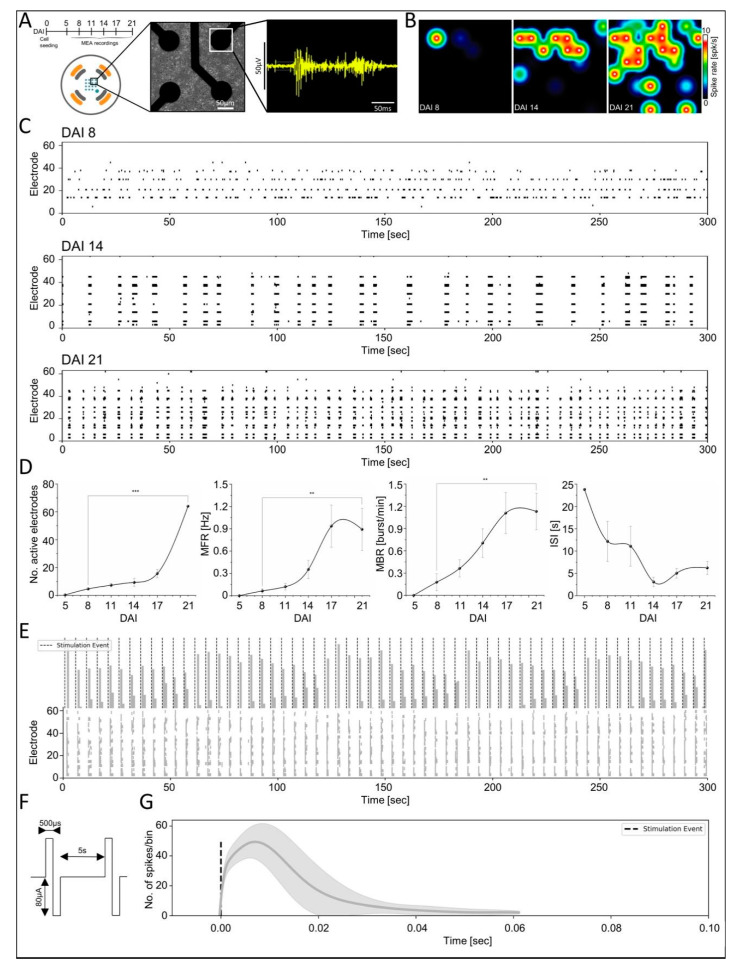
Network-wide characterization of spontaneous and evoked firing features. (**A**) Experimental timeline for extracellular MEA recordings (top). Representative single-well of 8 × 8 microelectrodes array and bright-field image displaying zoom in of 4 electrodes with seeded NPC-iNs neurons (bottom). Representative raw signal trace from a single electrode showing electrophysiological waveform properties as analogous to dissociated primary neuronal activity. (**B**) Activity heatmaps are displaying increased spatiotemporal activity based on spike/s during development. (**C**) Rastergrams of representative firing activity and synchronous bursting patterns from early to mature developmental phases. (**D**) Quantification of activity-dependent changes in parameters during development, including average number of active electrodes, MFR, MBR, and ISI as a function of DAI (*** *p* < 0.001, ** *p* < 0.01, ANOVA, N = 3). (**E**) Averaged-firing rate and raster plots of evoked responses following a series of stimulation events as a function of a 5 min time window. (**F**) A biphasic electrical stimulus delivered at 0.2 Hz with 500 µs per phase, and 160 µA peak-to-peak amplitude delivered from pair of electrodes distributed within array. (**G**) Exemplary averaged PSTH computed from all evoked activities corresponding to their stimulating site, with responses lasting 50 ms. Data are presented as mean ± s.e.m.

**Figure 4 cells-10-01894-f004:**
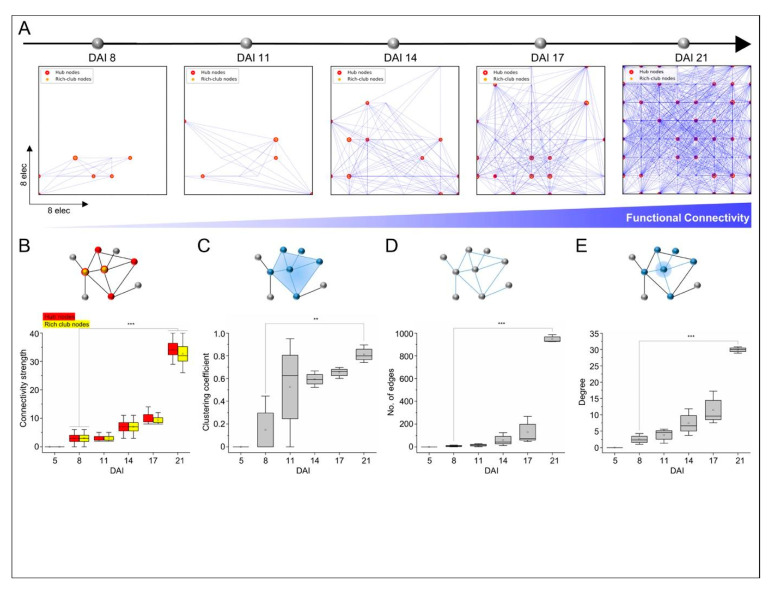
Emergence of functional connectivity and network topology during NPC-iNs network development. (**A**) Maps of functional connectivity during developmental phases showing increased spatial interconnected firing electrodes. (**B**) These maps also display hub nodes that are significantly increasing spatially with development by receiving communication through links from other nodes. Those node connections’ strength is also assessed with rich-club characterization that is significantly increasing during development (*p* < 0.001, ANOVA). (**C**–**E**) All graph topological metrics in matured NPC-iNs networks display significantly higher values than early developed networks (*p* < 0.01, 0.001, ANOVA, N = 3).

## Data Availability

The data are available from the corresponding authors upon request.

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
