# Peer review of "Improved Method for Efficient Generation of Functional Neurons from Murine Neural Progenitor Cells"

_cells, 2021, doi:10.3390/cells10081894_

Round 1

Reviewer 1 Report

General comments

In the paper entitled ”Improved method for efficient generation of functional neurons from murine neural progenitor cells” the authors have addressed the important issue of establishing in vitro culture conditions suitable to generate neurons from NSPCs and studying their functionality,.

The manuscript is easily readable, well written and provide additional evidence that NSPCs represent a suitable cellular source to generate electrophysiological functional neurons, able to form in vitro connections; limiting the use of primary neuronal cultures is also relevant in respect to human studies, because of the limited availability and accessibility of brain tissue.

Although the cellular system has been well characterized at functional level, this reviewer believes that the manuscript would benefit from some additional experiments.

Specific comments

Lane 64: “However, the proportion of neuronal commitment from cultured NPCs is maximally around 30%”

Differentiation protocols are reported in the literature that allow to generate neurons from embryonic and adult cortical NSPCs with efficiency significantly higher than 30% (even without ngn2 overexpression). are reported in literature. These papers should be cited.

Lane 246-248: At_ 7 DAI,_E_GFP+ NPC-iNs expressed the neuronal markers MAP2 246 and NeuN (100 ± 0 % within 234 GFP+ cells; and 98.5 ± 3 % within 336 GFP+ cells, respectively) (Figure 1C, D).

It should be mentioned what is the percentage of neurons in the cultures. Neither quantification versus total cell number nor representative images including nuclear staining are shown, making the real efficiency of the differentiation protocols unclear.

Furthermore, what is the identity of the GFP negative cells? Do they all differentiate into astrocyte-like cells? What is the percentage of GFAP positive cells? Are there GFP negative cells undifferentiated or differentiated into other types? What is their percentage?

These data could be relevant in biochemical or molecular assays perspective.

Figure 1C: V-glut staining is not convincing. It seems that the staining is mainly localized in the cell body/nucleus. A Higher quality image is required.

Lane 253: “The majority of 253 NPC-iNs expressed VGLUT1 (99.7 ± 0.5 %), suggesting that NPC-iNs are primarily glutamatergic neurons”.

Cortical NSPCs generate glutamatergic neurons in vivo but several papers have shown that NSPC regional identity is only partially retained after in vitro expansion and NSPCs mainly differentiate into gaba-ergic neurons. Although ngn2 overespression might preserve the NSPC capability to produce glutamatergic neurons. I would suggest to strengthen the data to confirm their glutamatergic identity and include staining for GABA-ergic neuronal markers

Related to the point raised above, it is unclear why the authors used Bicuculine that is a GABA receptor antagonist. Furthermore the activation of IEG upon Bicuculine treatment would indicate that the inhibitory neurons are present.

How was the dose of Doxycycline chosen? Was the doxycycline toxicity tested on a control cell line (not expressing ngn2)?

Reviewer 2 Report

In this study the authors have described a method for obtaining neurons from murine neural progenitor cells. They combined cell transformation with a Ngn2-EGF lentivirus with an exposition to DOX, Forskolin, Rock inhibitor, FBS and CHIR 99021, improving the standard method to obtain cultures of neurons derived from neural progenitor cells. They demonstrated through different techniques the physiological functionality of these neurons in vitro. Furthermore, they defined the best timing to use these cells depending on the nature of the experiment to perform.
The authors did a great job on the figures of the manuscript. The manuscript is well written and concise with clear conclusions.
They described meticulously their method in order to make it easily reproducible.

The experiments are successful and well chosen, and results are concise and robust. 

However, I would like to suggest some minor comments for their consideration:

The authors have described a method to obtain neurons from NPCs transfecting with a Ngn2 lentivirus and using DOX. I consider that it is important to mention in the introduction that these methods are classically used independently, because The authors mention different biological approaches to culture neurons.

In the discussion, it is important to discuss the novelty and advantages of their method compared to methods described previously in similar papers, both in human NPCs (S.-M. Ho et al., Methods (2015), http://dx.doi.org/10.1016/j.ymeth.2015.11.019) and in rodent NPCs (C.-H. Park et al. / FEBS Letters 582 (2008) 537–542).

Reviewer 3 Report

Soni et al. describe the development of a neuronal culture from expandable mouse NPCs based on transcription factor programming, to offer an in vitro resource for studies of neuronal physiology and plasticity at an individual neurons or network scale. The method relies on a Dox-induced transient expression of Ngn2 and results in the generation of morphologically mature neurons that respond to pharmacological silencing/activation. Ultimately these form neuronal networks that acquire synchrony over time in firing rate. The authors go on to further characterize the network features as the culture progresses (hubs, edges, connectivity strength) showing that these become gradually more complex and mature.

The manuscript is well structured and explores an interesting alternative to the standard primary culture from the mouse embryo. I have only a few minor comments:

  1. To the introduction (l. 60-70): The authors refer to the need of a neuron-enriched culture for neuroscience studies. However, their culture is also composed of glial cells (Fig.1C) and it remains unclear to which extent. If neuronal enrichment in the described method is not improved this aspect should be downplayed. If it is, the article would benefit from detailing those proportions (neuron/glia).
  2. To the Methods:
    1. Please specify which gestational day was used to perform the NPC culture and provide the right reference. Ref 16 provides a protocol to obtain NSC/NPC from adult mice.
    2. Please insert the virus details (titer, volume applied).
    3. The “n” is not always provided, please add where missing (Fig.4, FigS1, FigS2).
    4. How was the cell survival assessed? The method seems to be missing.
  1. Results/Discussion
    1. The authors suggest that the cell death detected in the first 7days after transient Dox treatment occurs as consequence of prolonged NgN2 overexpression during neuronal differentiation. However, Dox-induced toxicity cannot be formally excluded without the appropriate control (Dox-treated cells, previously infected with the control LV (without Ngn2)). Dox-induced effects on cell metabolism and apoptosis are known to vary between (human) cell lines. The authors should add a control experiment or reference(s) to support their statement or otherwise rephrase in order to accommodate this possibility.

Round 2

Reviewer 1 Report

The authors made a clear, thorough, and extremely detailed revision addressing all my concern. According to this reviewer, the authors improved their manuscript and I have no additional comments.

Reviewer 3 Report

The authors addressed all my questions and greatly improved the manuscript  that now provides a better understanding of their in vitro system. I have no further concerns and recommend acceptance in the present form.